# Incorporation of NiO into SiO_2_, TiO_2_, Al_2_O_3_, and Na_4.2_Ca_2.8_(Si_6_O_18_) Matrices: Medium Effect on the Optical Properties and Catalytic Degradation of Methylene Blue

**DOI:** 10.3390/nano10122470

**Published:** 2020-12-10

**Authors:** Carlos Diaz, María L. Valenzuela, Olga Cifuentes-Vaca, Marjorie Segovia, Miguel A. Laguna-Bercero

**Affiliations:** 1Departamento de Química, Facultad de Química, Universidad de Chile, La Palmeras 3425, Nuñoa, Casilla 653, 7800003 Santiago de Chile, Chile; msmonrroy@gmail.com; 2Inorganic Chemistry and Molecular Material Center, Facultad de Ingeniería, Instituto de Ciencias Químicas Aplicadas, Universidad Autónoma de Chile, Av. El Llano Subercaseaux 2801, San Miguel, 8910060 Santiago de Chile, Chile; maria.valenzuela@uautonoma.cl; 3Departamento Ciencias Químicas, Facultad de Ciencias Exactas, Universidad Andres Bello, Sede Concepción, Autopista Concepción-Talcahuano, 7100 Talcahuano, Chile; olcifuen@hotmail.com; 4Instituto de Nanociencia y Materiales de Aragón (INMA), CSIC-Universidad de Zaragoza, 50009 Zaragoza, Spain

**Keywords:** nickel oxide, photocatalysis, chitosan, polyvinylpyrrolidone, optical properties

## Abstract

The medium effect of the optical and catalytic degradation of methylene blue was studied in the NiO/SiO_2_, NiO/TiO_2_, NiO/Al_2_O_3_, and NiO/Na_4.2_Ca_2.8_(Si_6_O_18_) composites, which were prepared by a solid-state method. The new composites were characterized by XRD (X-ray diffraction of powder), SEM/EDS, TEM, and HR-TEM. The size of the NiO nanoparticles obtained from the PSP-4-PVP (polyvinylpyrrolidone) precursors inside the different matrices follow the order of SiO_2_ > TiO_2_ > Al_2_O_3_. However, NiO nanoparticles obtained from the chitosan precursor does not present an effect on the particle size. It was found that the medium effect of the matrices (SiO_2_, TiO_2_, Al_2_O_3_, and Na_4.2_Ca_2.8_(Si_6_O_18_)) on the photocatalytic methylene blue degradation, can be described as a specific interaction of the NiO material acting as a semiconductor with the M_x_O_y_ materials through a possible p-n junction. The highest catalytic activity was found for the TiO_2_ and glass composites where a favorable p-n junction was formed. The isolating character of Al_2_O_3_ and SiO_2_ and their non-semiconductor behavior preclude this interaction to form a p-n junction, and thus a lower catalytic activity. NiO/SiO_2_ and NiO/Na_4.2_Ca_2.8_(Si_6_O_18_) showed a similar photocatalytic behavior. On the other hand, the effect of the matrix on the optical properties for the NiO/SiO_2_, NiO/TiO_2_, NiO/Al_2_O_3_, and NiO/Na_4.2_Ca_2.8_(Si_6_O_18_) composites can be described by the different dielectric constants of the SiO_2_, TiO_2_, Al_2_O_3_, Na_4.2_Ca_2.8_(Si_6_O_18_) matrices. The maxima absorption of the composites (λ_max_) exhibit a direct relationship with the dielectric constants, while their semiconductor bandgap (E_g_) present an inverse relationship with the dielectric constants. A direct relationship between λ_max_ and E_g_ was found from these correlations. The effect of the polymer precursor on the particle size can explain some deviations from this relationship, as the correlation between the particle size and absorption is well known. Finally, the NiO/Na_4.2_Ca_2.8_(Si_6_O_18_) composite was reported in this work for the first time.

## 1. Introduction

Metal oxide nanoparticles are widely used in many applications such as coatings, catalysis, electrode materials, or sensors [1]. It is important to remark that their physical and chemical properties are strongly influenced by their agglomeration [2]. In this sense, it is well known that the incorporation of metal oxides onto inert support materials with high surface areas could help prevent particle agglomeration and also improve their reactivity and stability [3,4].

NiO is a p-type semiconductor with E_G_ = 3.5 eV presenting multiple practical applications [4,5,6]. However, their band gap can be modified by doping with other metal oxide semiconductors, and thus changing their photocatalytic properties [5,6]. NiO has been widely used in catalysis, battery cathodes, fuel cell electrodes, electrochromic films, electrochemical supercapacitors, or magnetic materials [4,5,6]. In this sense, Bonomo et al. [7] recently reported on the electrochemical and opto-electrochemical properties of nanostructured NiO for photoconversion applications. Although these applications are determined by their band-gap, which depend on the environment [8,9], no systematic studies have been reported regarding the effect of the medium on the band-gap behavior [10,11,12]. In this sense, it is well known that the dielectric medium affects the optical properties of nanoparticles, as previously observed for Au and Ag systems [10]. The optical properties of Au nanoparticles embedded into TiO_2_, ZrO_2_, and Al_2_O_3_ have been also studied qualitatively [10]. In addition, the effect of SiO_2_, TiO_2_, and ZrO_2_ supports was recently analyzed showing that MoO_3_/SiO_2_ is the most efficient epoxidation catalyst [12].

The Na_4.2_Ca_2.8_(Si_6_O_18_) compound (combeite) is a crystalline phase normally obtained from the fusion of precursor Na_2_O⋅CaO⋅SiO_2_ glasses [13,14,15]. In this sense, there are no reported metal oxides using Na_4.2_Ca_2.8_(Si_6_O_18_) as a solid matrix.

In previous works, we have reported a method to prepare metal and metal oxide nanostructured materials from a thermal treatment of the Chitosan (ML_n_)_x_ and PS-co-4-PVP (ML_n_)_x_ macromolecular complexes [16,17,18]. The method consists of two steps: (1) Formation of both macromolecular complexes by a solvent assisted reaction between the respective polymer and the metallic salt; and (2) a thermal process of the solid under air atmosphere.

The M° and M_x_O_y_ nanostructures can be easily incorporated into SiO_2_ matrices using a similar approach by different thermal treatments of the solid-state precursors: Chitosan (ML_n_)_x/_/SiO_2_ and PS-co-4-PVP (ML_n_)_x/_/SiO_2_ affording M_x_O_Y_//SiO_2_ composites [19,20]. This method can be also used to prepare NiO//M_x_O_y_ composites using SiO_2_, TiO_2_, Al_2_O_3_, or Na_4.2_Ca_2.8_(Si_6_O_18_) matrices. Although a few methods were proposed to prepare NiO/SiO_2_ [4,21,22], NiO/TiO_2_ [6,23,24,25], NiO/Al_2_O_3_ [26,27,28,29] composites, none of them is as general and simple as the one described here. As for the Na_4.2_Ca_2.8_(Si_6_O_18_) case, although this particular composition has not been reported, similar nickel oxide doped with silica matrices have been successfully synthesized via a sol–gel process [30]. Furthermore, this solid state method has been used for other systems [31]. A summary of the proposed fabrication route is shown in Figure 1 [13].

In addition, the effect of the different matrices on the optical properties will also be studied and discussed.

## 2. Materials and Methods

NiCl_2_·6H_2_O, tetraethyl orthosilicate (TEOS), chitosan, poly(styrene-*co*-4-vinilpyridine) PS-co-4-PVP, ethyl alcohol, acetic acid, and dichloromethane were supplied from Sigma-Aldrich and were used as received.

### 2.1. Preparation of the NiO/SiO_2_, NiO/TiO_2_, NiO/Al_2_O_3_ Composites

SiO_2_ was prepared according to the literature procedures [19,20]. Briefly, tetraethoxysilane (TEOS), ethanol, and acetic acid were mixed in a molar ratio of 1:4:4 with water (nanopure milli-Q), and added over the dichloromethane solution of the previously prepared chitosan (NiCl_2_·6H_2_O)_x_ and PS-co-4-PVP (NiCl_2_·6H_2_O)_x_. The mixture was stirred for 3 days. The obtained gel was dried at 100 °C under a vacuum. The chitosan (NiCl_2_·6H_2_O)_x_//SiO_2_ and PS-co-4-PVP (NiCl_2_·6H_2_O)_x_//SiO_2_ precursors were finally calcined at 800 °C for 2 h under air.

### 2.2. Preparation of the Chitosan (NiCl_2_·6H_2_O)_x/_/TiO_2_ and PS-co-4-PVP (NiCl_2_)_x/_/TiO_2_ Precursors

TiO_2_ was prepared according to the literature procedures [19,20]. Briefly, titanium tetra-isopropoxide (Ti(OC_3_H_7_)_4_, TTIP) ethanol and acetic acid were mixed in a molar ratio of 1:4:4 with water (nanopure milli-Q), and added over the dichloromethane solution of the previously prepared chitosan (NiCl_2_·6H_2_O)_x_ and PS-co-4-PVP (NiCl_2_·6H_2_O)_x_. The mixture was stirred for 3 days. The obtained gel was dried at 100 °C under a vacuum. The solid chitosan (NiCl_2_·6H_2_O)_x/_/TiO_2_ and PS-co-4-PVP (NiCl_2_·6H_2_O)_x_//TiO_2_ precursors were calcined at 800 °C for 2 h under air.

### 2.3. Preparation of the Chitosan (NiCl_2_·6H_2_O)_x/_/Al_2_O_3_ and PS-co-4-PVP (NiCl_2_)_x/_/Al_2_O_3_ Precursors

Al_2_O_3_ was prepared according to the literature procedures [27,28,29,30]. Briefly, AlCl_3,_ ethanol, and acetic acid were mixed in a molar ratio of 1:4:4 with water (nanopure milli-Q), and added over the dichloromethane solution of the previously prepared chitosan (NiCl_2_·6H_2_O)_x_ and PS-co-4-PVP (NiCl_2_·6H_2_O)_x_. The mixture was stirred for 3 days. The obtained gel was dried at 100 °C under a vacuum. The solid chitosan (NiCl_2_·6H_2_O)_x_//Al_2_O_3_ and PS-co-4-PVP (NiCl_2_·6H_2_O)_x_//Al_2_O_3_ precursors were calcined at 800 °C for 2 h under air.

### 2.4. Preparation of the Precursors: Chitosan (NiCl_2_·6H_2_O)_x_//NiO/Na_4.2_Ca_2.8_(Si_6_O_18_) and PS-co-4-PVP (NiCl_2_)_x_//NiO/Na_4.2_Ca_2.8_(Si_6_O_18_)

The compounds were prepared according to the literature procedures [28]. Briefly, tetraethoxysilane (TEOS), ethanol, and acetic acid were mixed in a molar ratio of 1:4:4 with water (nanopure milli-Q), then Na_2_O, CaO, and SiO_2_ solids (in mol% of 14:1.5:73) were added over the dichloromethane solution of the previously prepared chitosan (NiCl_2_·6H_2_O)_x_ and PS-co-4-PVP (NiCl_2_·6H_2_O)_x_. The mixture was stirred for 3 days. The obtained gel was dried at 100 °C under a vacuum. The solid chitosan (NiCl_2_·6H_2_O)_x_//Na_2_O CaO SiO_2_ and PS-co-4-PVP (NiCl_2_·6H_2_O)_x_//Na_2_O CaO SiO_2_ precursors were calcined at 800 °C for 2 h under air.

The coordination of the polymer was confirmed by IR analysis, as the broad ν(OH)+ ν(NH) band observed at 3448 cm^−1^ for free chitosan becomes unfolded upon coordination, shifting in the range of 3345–3393 cm^−1^. On the other hand, the ν(py) band is shifting to high frequencies upon coordination [16,17,18].

Finally, polymer-metal complexes were placed into a box furnace (lab tech) using a pyrolysis temperature of 180 °C for the precursor complexes and 800 °C for the polymer complexes. Additional experimental conditions are summarized in Table 1.

### 2.5. Characterization

IR spectra were recorded with a FT-IR Jasco 4600 spectrophotometer (Jasco Inc., Easton, MD, USA). Scanning electron microscopy (SEM) was performed on a JEOL 5410 scanning electron microscope (JEOL Ltd., Tokyo, Japan). Elemental microanalysis was performed by energy dispersive X-ray (EDS) analysis using a NORAN Instrument micro-probe attached to the SEM (Thermo Scientific, Waltham, MA, USA). High-resolution transmission electron microscopy (HR-TEM) was performed using a JEOL 2000FX TEM microscope (JEOL Ltd., Tokyo, Japan)at 200 kV to characterize the average particle size, distribution, and elemental and crystal composition. EDS analysis was performed in individual particles in order to discriminate NiO from the matrix. Average particle sizes were calculated using the Digital Micrograph software (Gatan, Inc., Pleasanton, CA, US). Methylene blue (MB) was used as a model compound to test the photocatalytic properties at 655 nm under UV-Vis illumination (Shimadzu UV-2600 spectrophotometer, Shimadzu Coorporation, Kyoto, Japan) using a xenon lamp (150 W) positioned 20 cm away from the photoreactor in a 330–680 nm range at room temperature, to avoid the self-degradation and thermal catalytic effects of cationic dye. Suspensions were stirred in the dark for 60 min to establish an adsorption/desorption equilibrium, after which the photocatalytic discoloration of MB was initiated.

## 3. Results and Discussion

### 3.1. Composite NiO/SiO_2_

The X-ray diffraction pattern of the as-synthesized NiO/SiO_2_ composite for the material from the chitosan precursor is shown in Figure 2a. All the reflection peaks of the XRD pattern can be indexed to NiO and SiO_2_ phases [19] (JPDS no. 03-065-2901 for NiO and JPDS no. 01-088-1535 for SiO_2_). The broad feature appearing at 22° corresponds to amorphous silica [19]. Similar X-ray diffraction patterns for NiO from the PVP precursor were obtained.

The SEM analysis (Figure 2b) shows irregular particle agglomerates, as typically observed from the preparation of nanoparticles using the solid-state thermal route [30]. From the TEM analysis, the agglomeration of NiO nanoparticles embedded into a mesh of SiO_2_ can be observed in Figure 2c, where these agglomerates are composed of fused NiO nanoparticles. The size of these nanoparticles are in the range of 14 nm with a mean size of 25 nm (Figure 2c). Detailed HR-TEM images in Figure 2e,f show a homogeneous dispersion of NiO over the silica network. However, it was not possible to acquire high resolution images in order to study the interfaces between NiO and the different matrices. In any case, as also confirmed by SEM-EDS mapping (Figure 2g), there is a uniform distribution of NiO and SiO_2_ particles. Similar results were observed for NiO obtained from the PVP precursor (see Appendix A). The only difference is that NiO particles are bigger in size ca. 100 nm.

### 3.2. NiO/TiO_2_

Figure 3 shows the XRD pattern of the NiO/TiO_2_ nanocomposite from the chitosan precursor, where the anatase phase and NiO are observed as single phases. Using this method, the pure TiO_2_ anatase phase was obtained, in contrast with other solution methods, where a mixture of anatase and rutile in the NiO/TiO_2_ composite was obtained [22]. The NiO/TiO_2_ composite shows a “cotton” type morphology from the chitosan precursor (Figure 3b), whereas the morphology from the PVP precursor presents a more densified structure, as shown in Figure 3c. The SEM-EDS mapping, shown in Figure 2g, indicates an homogeneous distribution of NiO and TiO_2_. Similar results were obtained for the NiO/TiO_2_ from the PVP precursor (see Appendix A).

The TEM analysis (Figure 3d,e) presents a “spider web” TiO_2_ network where the NiO nucleates forming agglomerated nanoparticles. They present a mean particle size of 25 nm (Figure 2f). A similar TEM analysis was observed for NiO/TiO_2_ obtained from the PVP precursor (Figure 3b and Appendix A).

### 3.3. NiO/Al_2_O_3_

Figure 4a shows the XRD pattern of the NiO/Al_2_O_3_ composite from the chitosan precursor where the corresponding peaks of γ-Al_2_O_3_ and NiO can be observed.

The effect of the polymer template on the morphology can be observed in Figure 4b,c. The chitosan precursor induces a “cotton” type morphology, while the PVP precursor also combines dense and irregular zones. Figure 4f shows an elemental mapping image demonstrating that NiO is well dispersed inside Al_2_O_3_. A complete characterization is shown in Appendix A.

As observed for the NiO/TiO_2_ system, the TEM analysis (Figure 4e) shows a “spider web” network of Al_2_O_3_ where the NiO nucleates form agglomerates. The histogram (Appendix A) shows a particle mean size of 17 nm. The HRTEM image of the NiO/Al_2_O_3_ from the PVP precursor is shown in Appendix A, Figure 3c, where it can be observed that the medium particle size is 32 nm.

### 3.4. NiO/Na_4.2_Ca_2.8_(Si_6_O_18_)

The XRD pattern of the NiO/Na_4.2_Ca_2.8_(Si_6_O_18_) composite prepared from the chitosan precursor indicates the formation of NiO inside the glass Na_4.2_Ca_2.8_(Si_6_O_18_) (see Figure 5a). The XRD pattern is in agreement with those reported in the literature [13,14,15]. The observed morphology is similar to the one previously reported [13,14,15] (see Figure 5b,c), also presenting a uniform distribution of NiO inside the Na_4.2_Ca_2.8_(Si_6_O_18_) (Figure 5d). Similar conclusions can be deduced for the PVP precursor (see Appendix A).

A summary of the medium particle sizes for NiO included into the different matrices is presented in Table 2, where the effect of the matrix and that of the polymer precursors on the final particle sizes can be observed.

The nanoparticle size of NiO obtained from the PVP precursor inside the matrices follow the order of SiO_2_ > TiO_2_ > Al_2_O_3_, while that for the NiO from the chitosan precursor does not present a significant effect on the nanoparticle size.

### 3.5. Photocatalytic Behavior

Although the main applied property of NiO is in the field of electrochemistry as Li-ion batteries [32] and supercapacitors applications, [33] its application as a photocatalytic activity toward organic dyes have also been suggested [34]. In any case, reports on the photocatalytic activity toward organic dyes using NiO/matrices are scarce. Yu et al. [6] found a higher photocatalytic activity for NiO/TiO_2_ than for pure NiO, towards the photodegradation of p-chlorophenol. Regarding the photocatalytic efficiency when using composites, important parameters to be considered include the formation of hierarchical porous structures, the dispersion of the catalytic semiconductor on the matrix surface, and the p-n junction in a NiO/M_x_O_y_ composite, where a new band gap will be formed with a most favorable value for the photodegradation chemical processes.

### 3.6. NiO

Methylene blue (MB) is extensively used as an organic dye in coloring paper, temporary hair colorant, dyeing cottons, and coating for paper stock [35]. The removal of this hazardous dye is considered as one of the growing requirements in recent years. The photocatalytic experiments were carried on the sample with definite dye concentration under dark conditions and UV irradiation. The band-gap of the NiO is 5.0 and 5.2 eV, when it is prepared from chitosan and PVP precursors, respectively. For the semiconductor metal oxides, their band gap value dictates their photocatalytic activity [35,36]. For this reason, the band gap of the C_3_–C_8_ composites was determined. These values are: 5.0, 5.2, and 5.4 eV for the NiO/SiO_2_, NiO/TiO_2_, NiO/Al_2_O_3_ composites, respectively, all obtained from the chitosan precursors. The values for the PVP precursor are: 5.5 eV, 5.2 eV for the NiO/SiO_2_, NiO/TiO_2_ composites, respectively. Those values do not change significantly, and are slightly higher than those reported previously, which can be due to their bigger particle sizes [34] (see Appendix A).

The changes in the absorption spectra of the MB aqueous solution exposed to UV light for various times in the presence of NiO are shown in Appendix A. The peak at 655 nm is characteristic of methylene blue and decreases with the irradiation time. Figure 6 shows the plot of time vs. concentration of methylene blue measured as C/C_o_ for NiO arising from both precursors, obtaining a catalytic efficiency of ~68% and ~71% of degradation in 5 h (see Figure 6c). Both degradation processes follow a zero order, as shown in Figure 6b,d. As previously mentioned, only a few photodegradation studies for NiO have been reported. For example, using 3 nm NiO nanoparticles [34] and NiO nanofibers [5], a moderated catalytic activity towards Rhodamine B was observed. In both cases, the degradation kinetic was zero order, which means that the rate of degradation does not depend on the MB concentration. This type of model is normally observed when the surface of the photocatalyst is saturated with the dye, so that the degradation rate remains relatively constant, depending only on the generation of photo-induced charges in the catalyst.

### 3.7. NiO in Matrices

The photocatalytic activity towards MB degradation for the NiO composite using different matrices is shown in Figure 7. The degradation rate of the NiO/TiO_2_ composite is shown for comparison. In any case, the photocatalytic activity of these NiO compounds is still far from the pure TiO_2_ standard phase [37]. For example, we have recently reported a 98% discoloration rate in only 25 min for TiO_2_ nanostructures using similar synthetic routes, and the degradation of commercial TiO_2_ (Degussa P25) is about 75% of MB under the same experimental conditions [38]. A representative plot of MB absorption at 655 nm vs. time is given in Appendix A. A summary of the kinetic degradation data is also displayed in Table 3.

As seen in Figure 6, the NiO from the chitosan precursor produces a higher activity than that arising from the PVP precursor. These results also apply for both SiO_2_ and TiO_2_ matrices. Interestingly, the most efficient photocatalytic activity was observed for the NiO/TiO_2_ composite with a 91% degradation of methylene blue in 5 h. This can be probably related with a matrix effect of SiO_2_, TiO_2_, and Al_2_O_3_.

Our results of catalytic degradation for the NiO//TiO_2_ composite (about 91%) are similar or slightly higher than those reported in the literature. Ahmed claimed 90% of catalytic degradation efficiency on the NiO//TiO_2_ composite prepared from titanium chloride and nickel acetylacetonate [39]. Faisal et al. obtained a similar catalytic degradation efficiency using an ultrasonication method [40]. Sim et al. reported 86% of the degradation efficiency using plasma enhanced chemical vapor deposition (PECVD) with hydro-oxygenated amorphous titanium dioxide obtained from titanium tetra-isopropoxide [Ti(OC3H7)4, TTIP] liquid as a precursor [41]. Finally, Chen et al. reported 86% catalytic degradation of MB using a method that involves incipient wet impregnation of the nickel oxide (NiO) nanoparticles over previously prepared TiO_2_ nanotubes [24].

It is suggested that for the most catalytically active TiO_2_ as the matrix, a p-n junction can be formed acting NiO as p-NiO and TiO_2_ as n-TiO_2_, see Appendix A, leading to a reduction of the recombination rate of photogenerated electron-hole pairs, which is known to enhance the photocatalytic activity of TiO_2_. A detailed description of the mechanism can be found on Appendix A. Therefore, it seems that the matrix is playing a crucial role for the NiO/TiO_2_ composite and in this case, the NiO acts as the matrix rather than an active semiconductor. On the other hand, the less efficient photocatalyst toward MB degradation arises probably from an insulating Al_2_O_3_ effect [28,42], which preclude the p-NiO behavior. This is in agreement with the observed photocatalytic decrease for the TiO_2_/SiO_2_ composite in comparison with pure TiO_2_. In the case of the NiO/SiO_2_ composite, the lower photocatalytic activity is probably a consequence of the high porous morphology which is induced by the SiO_2_ matrix. All the photodegradation processes of MB with NiO/SiO2, NiO/TiO_2_, NiO/Al_2_O_3_, and NiO/Na_4.2_Ca_2.8_(Si_6_O_18_) composites exhibited a zero order kinetic law, as shown in Appendix A.

### 3.8. Photocatalytic Activity of the NiO/Na_4.2_Ca_2.8_(Si_6_O_18_) Composite

The photocatalytic activity of the NiO/Na_4.2_Ca_2.8_(Si_6_O_18_) composite obtained from the chitosan precursor is shown in Figure 7 and the kinetic data is also shown in Table 2. It is observed that the photocatalytic activity is higher than that of NiO, NiO/SiO_2_, and NiO/Al_2_O_3_ but lower than of NiO/TiO_2_. It is concluded that the Na_4.2_Ca_2.8_(Si_6_O_18_) sample presents a similar behavior to those of the SiO_2_ sample.

### 3.9. Effect of the Matrices on λ_max_ and E_g_

Figure 8 shows the variation of both E_g_ and λ_max_ for the different matrices. The respective UV-Vis absorption spectra of the composites are shown in Appendix A. The band-gap values were estimated from these spectra using the Tauc procedure (Appendix A). Considering that the static dielectric constants (K) for the matrices are: SiO_2_ 3.9; TiO_2_ 80, and Al_2_O_3_ 8.8 [43], both E_g_ and λ_max_ could be related to the dielectric constant of the matrix. Unfortunately, there is no available data for Na_4.2_Ca_2.8_(Si_6_O_18_). The dependence of E_g_ with the dielectric constant ε is not totally understood, where several relationships have been previously found [43,44,45]. The shape of the experimental or theoretical expression depends, among others, on the type of materials. On the other hand, the relationship of λ_max_ with ε and the refractive index n is known for metallic nanoparticles [8]:(1)$$\lambda_{\max}\;\alpha \;\lambda_{p}\sqrt{2\varepsilon +1}≅\sqrt{2}\;\lambda_{p}\;n$$

However, the analogue relationship for metal oxides is not completely understood. The plot of λ_max_ for the NiO vs. the refractive index [46,47] for the SiO_2_, TiO_2_, and Al_2_O_3_ matrices shows an inverse and irregular relationship (see Appendix A). Then, according to Figure 8, the variations of E_g_ and λ_max_ can be explained by a physical effect of the medium reflected in their dielectric constant of the different matrices. A close inspection of Figure 8 suggests the presence of three linear trends. In Figure 8, it can be observed that λmax varied inversely with the properties of the matrices (i.e., dielectric constant for instance) for the NiO obtained from both polymers in an approximate linear behavior. The composite C_8_ having a “silica like” matrix does not follow this trend due to an unknown effect. Although the dependence of λmax with the dielectric or refractive index given by Equation (1) indicates a direct linear dependence, our results show an inverse linear trend. Then, the Equation (1) may not be valid for metallic oxides. A new equation is proposed (Equation (2), curve a in Figure 8), which could arise from the general trends for nanostructured metallic oxides. This is consistent with the fact observed in Appendix A, where an inverse relationship of λmax with n is shown.

In addition, Eg values vary in a direct or inverse way depending on the NiO polymer precursor (direct behavior for the chitosan; curve b, Equation (3) or inverse for the PVP precursor; curve c, Equation (4)). As previously mentioned, the dependence of E_g_ with the dielectric constant ε is not totally understood, and this is a matter of controversy in the literature. The inverse relationship (curve c) is in agreement with the results reported by Hervé and Vandamme [48], while the direct relationship (curve b) shows a similar trend to that shown by Kumar and Singh [49]. In any case, we do not have any clear explanation of the different dependencies of Eg with n and ε when using the different polymer precursors.

From the plot shown in Figure 8, the following equations can be established:λ_max_ = a/(ε,n); valid for NiO from chitosan and PVP(2)
Eg = b/(ε,n); valid for NiO from chitosan(3)
Eg = c/(ε,n); valid for NiO from PVP(4)

The following equations are then obtained by combining both expressions:Eg = ab/λ_max_; valid for NiO from chitosan(5)
Eg = cλ_max_/a; valid for NiO from PVP(6)

In agreement with these new expressions, we can explain the effect of the physical properties of the matrices on the band gap with the refraction index or the dielectric constant. The experimental data fits into these equations, as seen in Figure 9 plots d (Equation (5)) and e (Equation (6)). These equations describing the effect of the medium modulated by various solid matrices on the band gap and the maximum absorption could be valid for other nanostructured metal oxides included in solid matrices. In order to validate this, additional experiments with other systems are being carried out.

Experiments linking the band gap with the size and the maxima absorption of nanoparticles have been performed for other metal oxides such as ZnO [50,51], as well as for noble metal nanoparticles such as Au [52], Ag [32], and Pt [53]. However, there are no studies in the literature about the medium expressed by solid matrices on nanostructured metallic oxides.

## 4. Conclusions

NiO/SiO_2_, NiO/TiO_2_, NiO/Al_2_O_3_, and NiO/glass composites were satisfactorily prepared by a solid-state synthesis from the chitosan and PVP precursors. XRD, SEM/EDS, and HR-TEM were used to characterize the new formed composites. It was concluded that the nature of the precursor polymer influences the morphology, as well as the size of the obtained nanoparticles. The chitosan precursor induces the smallest NiO nanoparticles and also their respective nanocomposites. In addition, the nature of the matrix influences the NiO nanoparticle size, following the order of SiO_2_ > TiO_2_ > Al_2_O_3_ for the PVP precursor. However, no relationship on the particle size was observed for the NiO obtained from the chitosan precursor.

The efficiency on the photocatalytic activity depends on the formation of a p-n junction between NiO acting as p-NiO and the metal oxide matrix acting as n-metal oxide. TiO_2_ presents the most effective p-NiO//n-TiO_2_ junction. On the other hand, the optical parameters Eg and λ_max_ depends on the dielectric constant and the refractive index of the matrix medium in a manner which depends on the preparation procedure. The “silica like” Na_4.2_Ca_2.8_(Si_6_O_18_) matrix does not follow these correlations. New equations describing the effect of the physical properties (dielectric constant and the refractive index) are proposed, which could be used for other metal oxides included in solid matrices.

## Figures and Tables

**Figure 1 nanomaterials-10-02470-f001:**
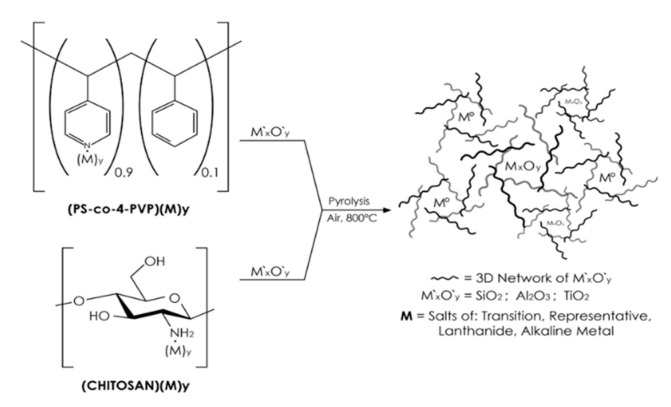
Schematic representation of the preparation method of metallic M° and metal oxides M_x_O_y_ nanoparticles inside M′_x_O′_y_ matrices.

**Figure 2 nanomaterials-10-02470-f002:**
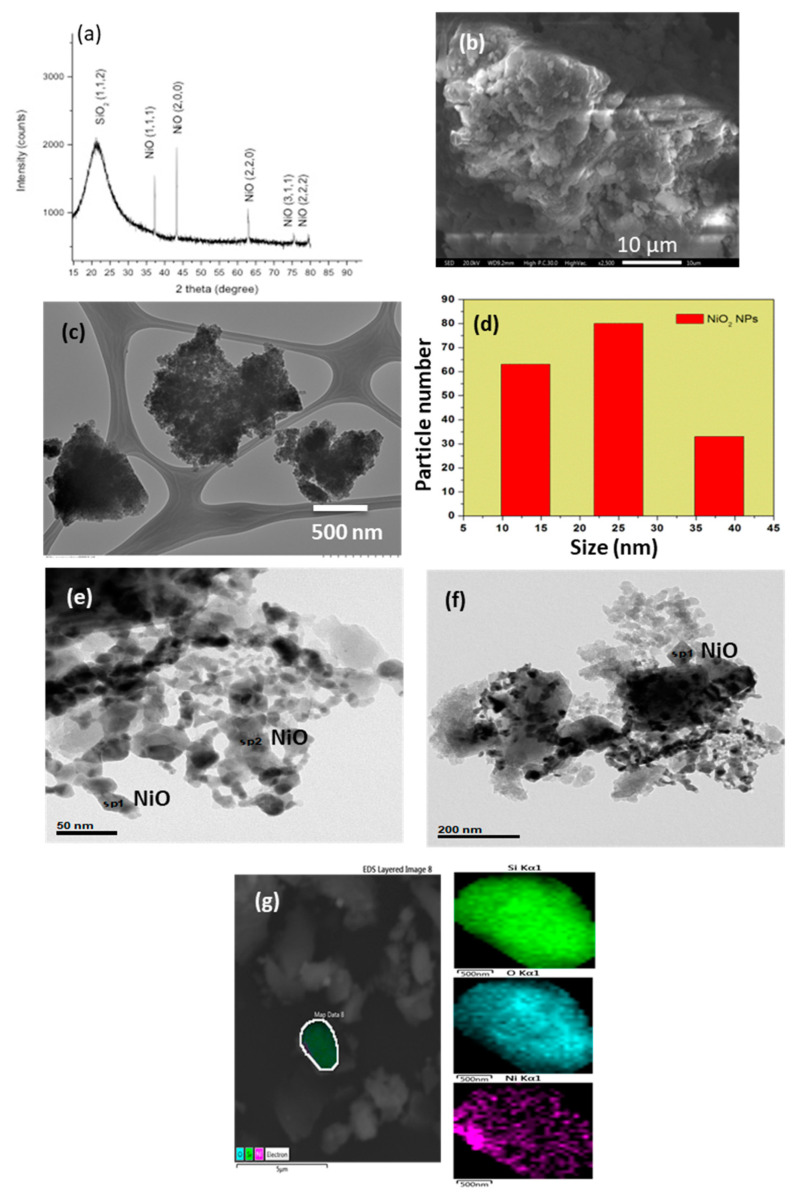
(**a**) XRD pattern; (**b**) SEM image; (**c**) TEM image; (**d**) particle histogram; (**e**,**f**) HRTEM images; and (**g**) SEM element mapping of the pyrolytic NiO compound obtained using the chitosan precursor.

**Figure 3 nanomaterials-10-02470-f003:**
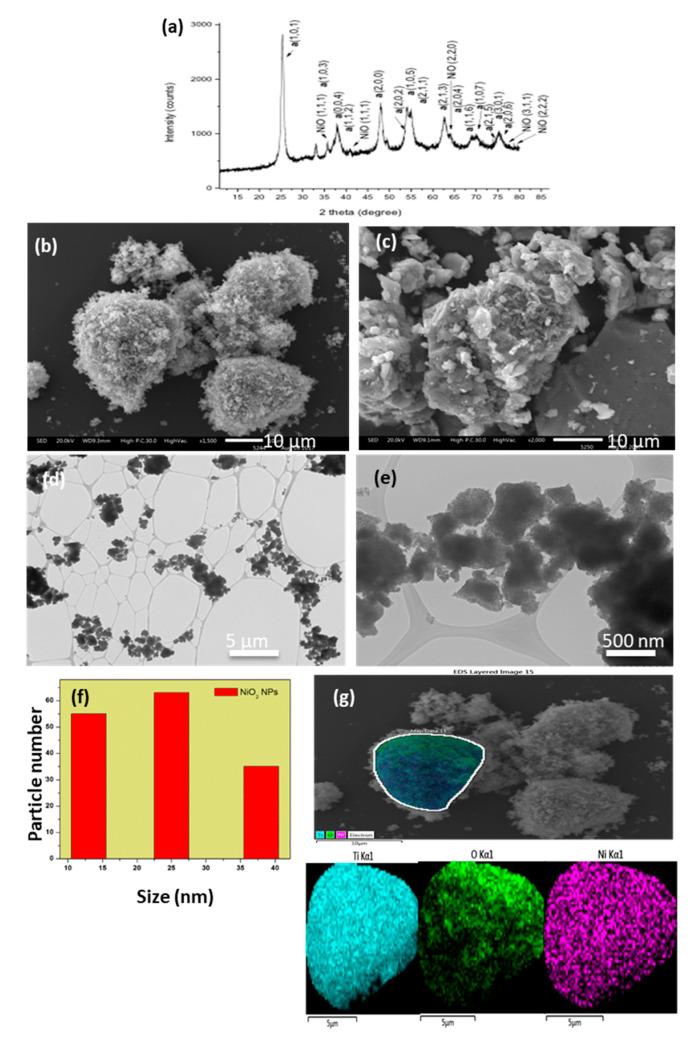
(**a**) XRD pattern; (**b**) SEM image of NiO from chitosan and (**c**) from PVP; (**d**,**e**) TEM images of NiO from chitosan and (**f**) their histogram; and (**g**) SEM mapping element for NiO from the chitosan precursor.

**Figure 4 nanomaterials-10-02470-f004:**
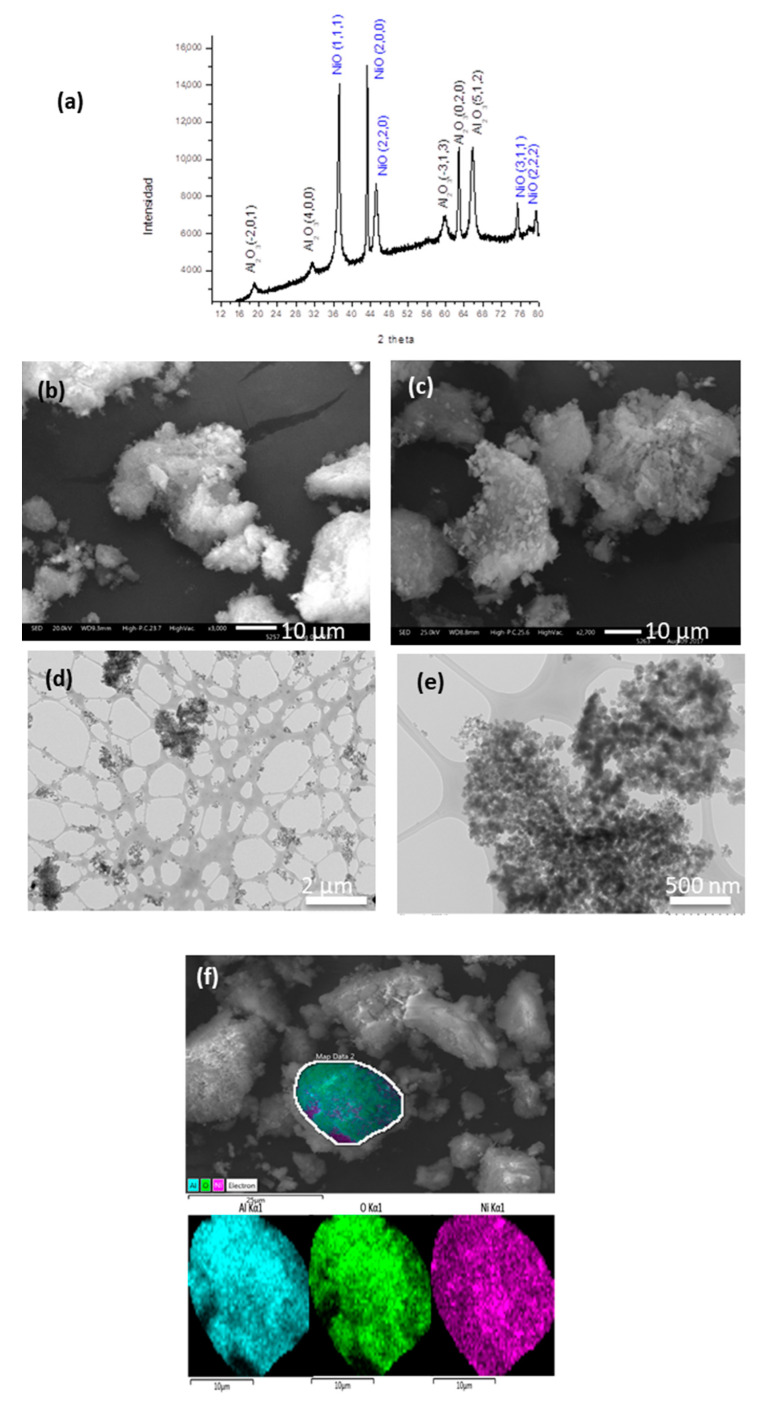
(**a**) XRD pattern of NiO/Al_2_O_3_ from the chitosan precursor; (**b**) SEM image of NiO from chitosan and (**c**) from PVP; (**d**) TEM image of NiO from chitosan and (**e**) from PVP; (**f**) EDS mapping of NiO from chitosan.

**Figure 5 nanomaterials-10-02470-f005:**
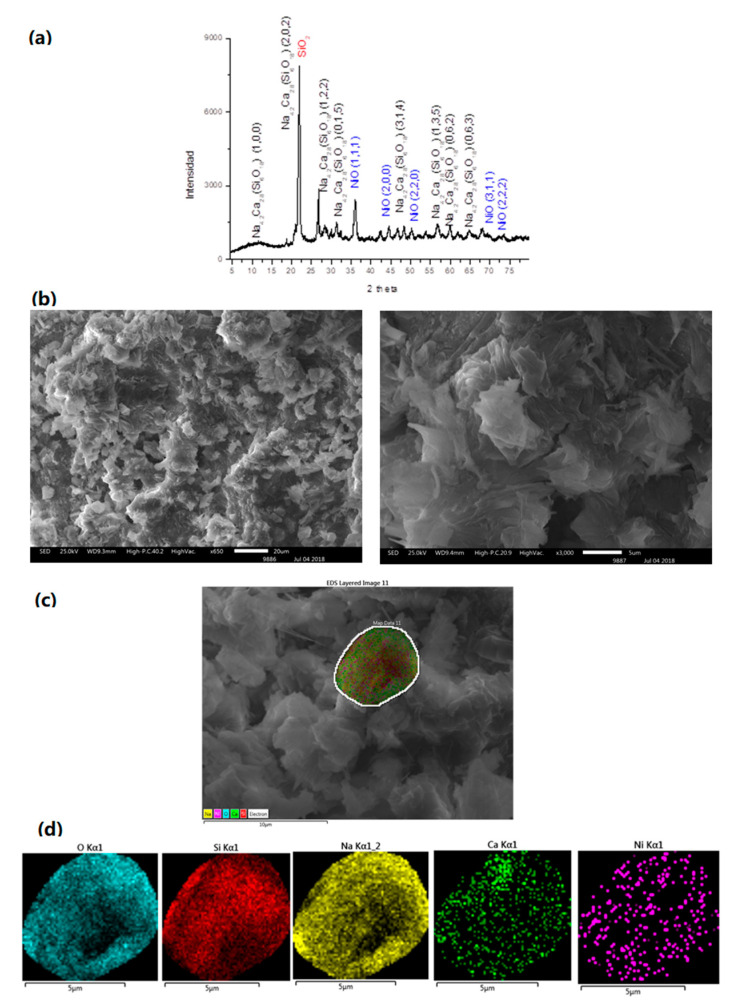
(**a**) XRD pattern of NiO inside Na_4.2_Ca_2.8_ (Si_6_O_18_); (**b**) and (**c**) SEM images; and (**d**) EDS mapping by an element of the composite NiO/Na_4.2_Ca_2.8_(Si_6_O_18_).

**Figure 6 nanomaterials-10-02470-f006:**
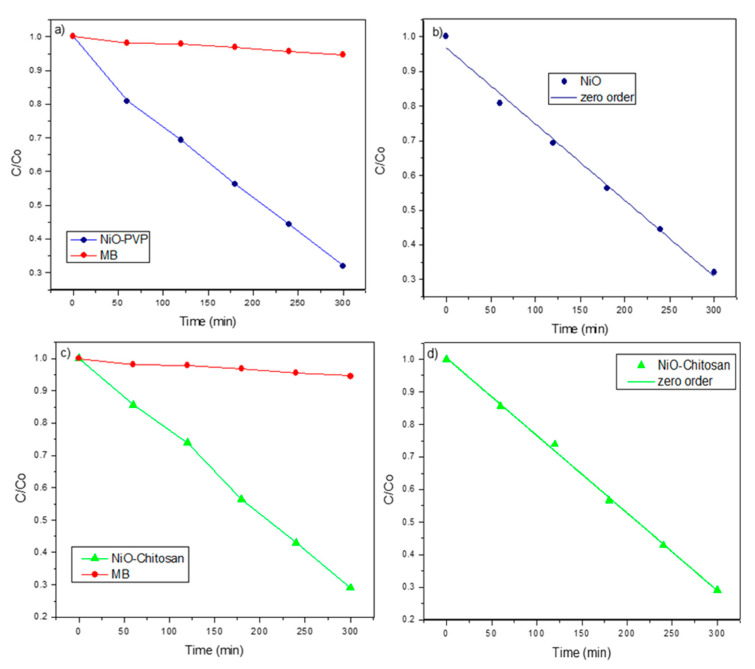
(**a**) Normalized concentration changing of methylene blue (MB) without the catalyst and in the presence of NiO from the PVP and (**b**) their zero order kinetic of degradation of MB, (**c**) changing of MB without the catalyst and in the presence of NiO from the chitosan and (**d**) their zero order kinetic of MB degradation.

**Figure 7 nanomaterials-10-02470-f007:**
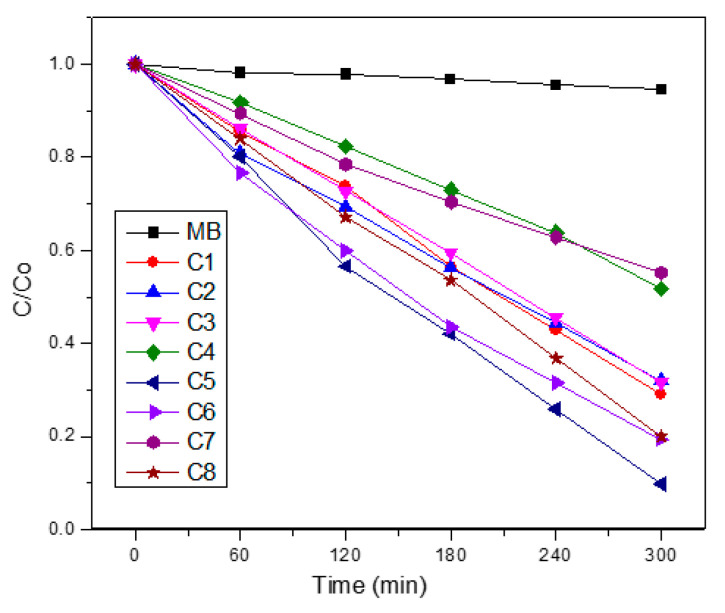
Photocatalytic behavior of NiO embedded in several matrices.

**Figure 8 nanomaterials-10-02470-f008:**
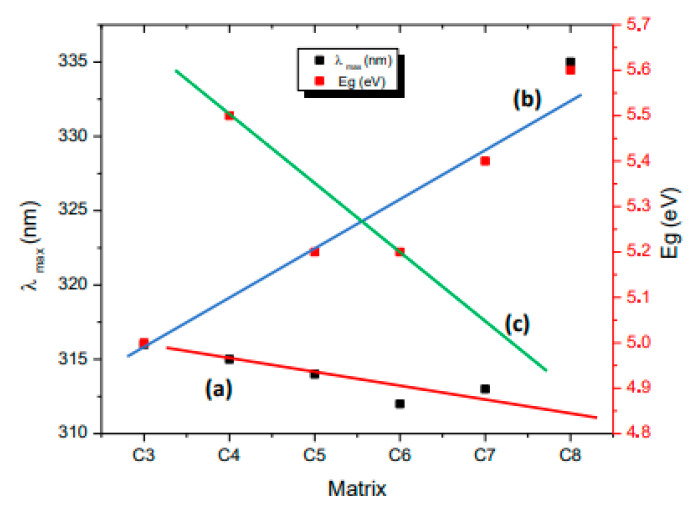
Variation of both E_g_ and λ_max_ with the different matrices.

**Figure 9 nanomaterials-10-02470-f009:**
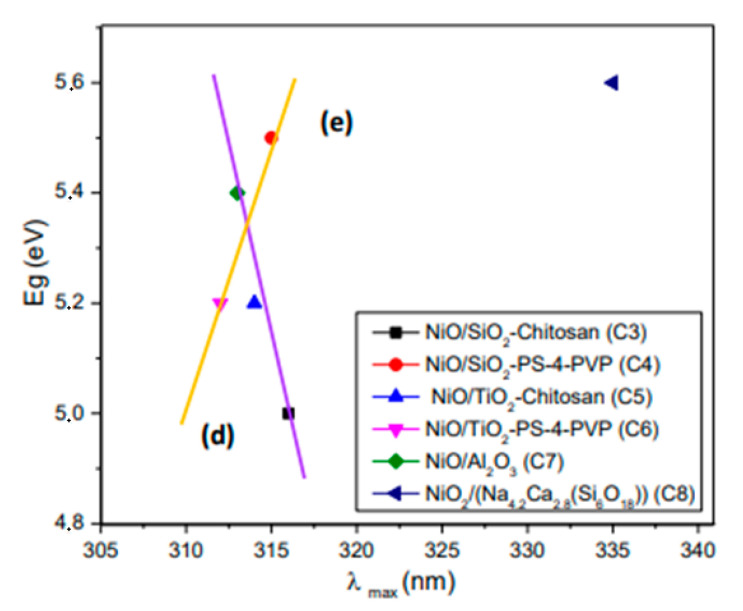
Relationship between E_g_ and λ_max_ for the composites the NiO/SiO_2,_ NiO/TiO_2_, NiO/Al_2_O_3_, and NiO/Na_4.2_Ca_2.8_(Si_6_O_18_).

**Table 1 nanomaterials-10-02470-t001:** Composition of the pyrolytic products from the respective precursors.

Precursor	Precursor Formula	Matrix	Composite	Composite Number
(1)	Chitosan·NiCl_2_ (chitosan)	-	NiO	C_1_
(2)	PSP-4-PVP·NiCl_2_ (PVP)	-	NiO	C_2_
(3)	Chitosan·NiCl_2_	SiO_2_	NiO/SiO_2_	C_3_
(4)	PSP-4-PVP·NiCl_2_	SiO_2_	NiO/SiO_2_	C_4_
(5)	Chitosan·NiCl_2_	TiO_2_	NiO/TiO_2_	C_5_
(6)	PSP-4-PVP·NiCl_2_	TiO_2_	NiO/TiO_2_	C_6_
(7)	Chitosan·NiCl_2_	Al_2_O_3_	NiO/Al_2_O_3_	C_7_
(8)	Chitosan·NiCl_2_	Na_4.2_Ca_2.8_(Si_6_O_18_)	NiO/Na_4.2_Ca_2.8_(Si_6_O_18_)	C_8_

**Table 2 nanomaterials-10-02470-t002:** Nanoparticle size for the composites.

Composite	Precursor Formula	Particle Size (nm)	Reference
NiO	Chitosan·NiCl_2_	>50	[17]
NiO	PSP-4-PVP·NiCl_2_	>50	[17]
NiO/SiO_2_	Chitosan·NiCl_2_	25	This work
NiO/SiO_2_	PSP-4-PVP·NiCl_2_	100	This work
NiO/TiO_2_	Chitosan·NiCl_2_	25	This work
NiO/TiO_2_	PSP-4-PVP·NiCl_2_	63	This work
NiO/Al_2_O_3_	Chitosan·NiCl_2_	30	This work
NiO/Al_2_O_3_	Chitosan·NiCl_2_	17	This work
NiO/Na_4.2_Ca_2.8_(Si_6_O_18_)	Chitosan·NiCl_2_	Not measured	This work

**Table 3 nanomaterials-10-02470-t003:** Kinetic data for the photodegradation process of MB with NiO and NiO/SiO_2,_ NiO/TiO_2_, NiO/Al_2_O_3_, and NiO/Na_4.2_Ca_2.8_(Si_6_O_18_) composites.

Photocatalyst	Apparent Photodegradation *	Discoloration Rate (%)	R^2^ Linear Fit (%)
NiO-CHITOSAN	2.4	71%	0.998
NiO-PS-4-PVP	2.2	68%	0.991
NiO/SiO_2_-CHITOSAN	2.3	69%	0.999
NiO/SiO_2_-PS-4-PVP	1.6	48%	0.996
NiO/TiO_2_ -CHITOSAN	2.9	91%	0.992
NiO/TiO_2_-PS-4-PVP	2.6	81%	0.980
NiO/Al_2_O_3_-CHITOSAN	1.5	45%	0.990
NiO/Na_4.2_Ca_2.8_(Si_6_O_18_)	2.6	75%	0.990

* Rate constant k (10^−3^ M·min^−1^).

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
