# Peer review of "Incorporation of NiO into SiO2, TiO2, Al2O3, and Na4.2Ca2.8(Si6O18) Matrices: Medium Effect on the Optical Properties and Catalytic Degradation of Methylene Blue"

_nanomaterials, 2020, doi:10.3390/nano10122470_

Round 1

Reviewer 1 Report

This manuscript reports on the fabrications of incorporation of NiO into different metal oxides for catalytic application. The content of this work is interesting, but requires revisions before its possible publication.

1. There are some grammatical errors in this article; the authors should do a thorough check. Major English revision is recommended.

2. All HRTEM images in this draft were not clear. Particularly, the interface between NiO and metal oxide should be studied.

3. The mechanism of the highest catalytic in NiO/TiO2 composites should be clarified.

4. The performance of degradation of methylene blue in NiO/TiO2 composites should be compared with literature reports.

5. How about the stability of degradation of methylene blue in NiO/TiO2 composites?

6. Most of old literature references should be updated. Some related new references should be read and can be cited in the Introduction section.

Ceram. Int. 2018, 44, 7047.

Catalysts 2018, 8, 575.

Applied Surface Science 2019, 488, 546.

J. Nano Res. 2019, 56, 152.

Applied Surface Science 2020, 511, 145548.

Author Response

This manuscript reports on the fabrications of incorporation of NiO into different metal oxides for catalytic application. The content of this work is interesting, but requires revisions before its possible publication.

  1. There are some grammatical errors in this article; the authors should do a thorough check. Major English revision is recommended.

English language was carefully revised.

  1. All HRTEM images in this draft were not clear. Particularly, the interface between NiO and metal oxide should be studied.

We acknowledge the reviewer for this comment, as our purpose was to study the interfaces in detail. However, we could not achieve this with precision due to the nature effect of the matrix, as it was not possible to acquire high resolution images for the studied samples. In this sense, a couple of sentences were added in the revised version (section 3.1.)

“Detailed HR-TEM images (2e and 2f) show a homogeneous dispersion of NiO over the silica network. However, it was not possible to acquire high resolution images in order to study the interfaces between NiO and the different matrices. In any case, as also confirmed by SEM-EDS mapping (figure 2g), there is a uniform distribution of NiO and SiO2 particles.”

  1. The mechanism of the highest catalytic in NiO/TiO2 composites should be clarified.

A detailed description of the mechanism can be found in figure S11.

S11 The photodegradation mechanism of NiO/TiO2 composites

When NiO-TiO2 nanocomposites are illuminated by visible light, the electrons of the Ni 3d sub-band are excited and transferred to the conduction band (see figure S11 in supplementary information). Accordingly, a high flux of free electrons is produced in the conduction band of TiO2. The photogenerated electrons in the conduction band of TiO2 reduce O2 species to O2-. This pathway is crucial to promote the photocatalytic efficiency in the MB oxidative decomposition. Then, the photoinduced holes (h+) in the valence bond of TiO2 are moved to the valence bond of NiO, generating a high flow of holes at the NiO interface. These holes react with H2O or OHions, producing extremely oxidative OH radicals, which proper oxidants in the photocatalytic oxidation process. These radicals react rapidly with methylene blue.

  1. The performance of degradation of methylene blue in NiO/TiO2 composites should be compared with literature reports.

The following text and references were included in the revised version.

Our results of catalytic degradation for the NiO//TiO2 composite (about 91%) are similar or slightly higher than those reported in the literature. Ahmed claimed 90 % of catalytic degradation efficiency on NiO//TiO2 composite prepared from titanium chloride and nickel acetylacetonate.1 Faisal et al. obtained similar catalytic degradation efficiency using an ultrasonication method.2  Sim et al. reported 86% of degradation efficiency using plasma enhanced chemical vapor deposition (PECVD) with hydro-oxygenated amorphous titanium dioxide obtained from titanium tetra-isopropoxide [Ti(OC3H7)4, TTIP] liquid as a precursor.3 Finally, Chen et al. reported 86% catalytic degradation of MB using a method that involve incipient wet impregnation of the nickel oxide (NiO) nanoparticles over previously prepared TiO2 nanotubes.4

References

1.- M.A. Ahmed, Synthesis and structural features of mesoporous NiO/TiO2 nanocomposites prepared by sol–gel method for photodegradation of methylene blue dye. Journal of Photochemistry and Photobiology A: Chemistry 238 (2012) 63–70.

2.- M. Faisal , Farid A. Harraza, , Adel A. Ismailc , Ahmed Mohamed El-Tonib , S.A. Al-Sayaria, Novel mesoporous NiO/TiO2 nanocomposites with enhanced photocatalytic activity under visible light illumination. Ceramic International, 2018, 44, 7047-7056.

3.- Lan Ching Sim, Kai Wern Ng, Shaliza Ibrahim, and Pichiah Saravanan, Preparation of Improved p-n Junction NiO/TiO2 Nanotubes for Solar-Energy-Driven Light Photocatalysis, International Journal of Photoenergy , Volume 2013, Article ID 659013, 10 pages http://dx.doi.org/10.1155/2013/659013.

4.- Jian-Zhi Chen  , Tai-Hong Chen  , Li-Wen Lai  , Pei-Yu Li  , Hua-Wen Liu  , Yi-You Hong  and Day-Shan Liu, Preparation and Characterization of Surface Photocatalytic Activity with NiO/TiO2 Nanocomposite Structure, Materials 2015, 8, 4273-4286.

  1. How about the stability of degradation of methylene blue in NiO/TiO2 composites?

Unfortunately, the stability of the composites was not studied in this work. However, as this is an important issue, it is currently being explored.

  1. Most of old literature references should be updated. Some related new references should be read and can be cited in the Introduction section.

The following references were now discussed and cited.

Ceram. Int. 2018, 44, 7047.

Applied Surface Science 2020, 511, 145548.

We have carefully read the other suggested references. Although they are interesting, they refer to TiO2 or NiO without forming composites. As a consequence, we have decided not to include them.

Catalysts 2018, 8, 575 (refers to TiO2)

Applied Surface Science 2019, 488, 546 (refers to TiO2 for CH3OH treatment)

  1. Nano Res. 2019, 56, 152. (refers to NiO)

Reviewer 2 Report

An efficient nano-catalyzed green synthesis, charachterization of substituted pyrazoles & study of their fluorescence charachteristics

The authors describe the optical and catalytic degradation of methylene blue in the NiO/SiO2, -TiO2, -Al2O3 and -Na4.2Ca2.8(Si6O18) composites prepared by a solid-state method. These nanpoparticles were characterized by XRD, SEM/EDS, TEM and HR-TEM. The medium effect of the matrices on the photocatalytic properties by the degradation of blue methylene, was due to a specific interaction of the NiO material as semiconductor with the metal oxide materials through a possible p-n junction. The highest catalytic activity was found for the NiO/TiO2 composite while NiO/SiO2 and NiO/Na4.2Ca2.8(Si6O18) showed a similar photocatalytic behavior. The optical parameters Eg and lmax of NiO depends on the dielectric constant and the refractive index of the matrix medium depending on the preparation from the chitosan and PS-co-PVP precursors.

The manuscript is suitable for publication after major revisions for the reasons specified below:

  1. Lines 63, 65, 72, 88, 92, 96-98, 100, 104, 106, 107, 110, 112, 115, 119, 121, 147, 200, 209, 210, 212, 217, 254, 307, 309, 370, 392, 393, 408-410, 445, 508, Table 1 and 2: the symbol typo “•” present in the text should be substituted with “∙”.

  1. Pg 3, Line 101: Please add this reference: Carlucci, H. Xu, B. F. Scremin, C. Giannini, T. Sibillano, E. Carlino, V. Videtta, G. Gigli, G. Ciccarella, Sci. Adv. Mater. 2014, 6, 1668-1675.

  1. Pg 3, Figure 1: the figure is unclear and M° in the caption is missing.

  1. Pg 4, Table 1: the yields and colors, indicated in the caption, are not reported.

  1. Pg 6, Figure 2: the scale is not formatted appropriately and in figures (e) and (f) is not reported. In figure (c) the caption of the axes is cut.

  1. Pg 8, Figure 3: the scale is not formatted appropriately and in figure (b) is missing. In figure (f) the caption of the y axis is cut.

  1. Pg 10, Figure 4: the scale is not formatted appropriately.

  1. Pg 11, 12, Figure 5: the scale is not formatted appropriately.

  1. Pg 13, Figure 6: the graphics seem to be pasted as an image and the resolution is low.

  1. References: please check carefully the numbering of the references (for example pg 12, lines 344-346).

  1. SI, Pg 7, Figure 5: the figure (B) is cut.

Author Response

An efficient nano-catalyzed green synthesis, charachterization of substituted pyrazoles & study of their fluorescence charachteristics

The authors describe the optical and catalytic degradation of methylene blue in the NiO/SiO2, -TiO2, -Al2O3 and -Na4.2Ca2.8(Si6O18) composites prepared by a solid-state method. These nanpoparticles were characterized by XRD, SEM/EDS, TEM and HR-TEM. The medium effect of the matrices on the photocatalytic properties by the degradation of blue methylene, was due to a specific interaction of the NiO material as semiconductor with the metal oxide materials through a possible p-n junction. The highest catalytic activity was found for the NiO/TiO2 composite while NiO/SiO2 and NiO/Na4.2Ca2.8(Si6O18) showed a similar photocatalytic behavior. The optical parameters Eg and lmax of NiO depends on the dielectric constant and the refractive index of the matrix medium depending on the preparation from the chitosan and PS-co-PVP precursors.

The manuscript is suitable for publication after major revisions for the reasons specified below:

  • Lines 63, 65, 72, 88, 92, 96-98, 100, 104, 106, 107, 110, 112, 115, 119, 121, 147, 200, 209, 210, 212, 217, 254, 307, 309, 370, 392, 393, 408-410, 445, 508, Table 1 and 2: the symbol typo “•” present in the text should be substituted with “∙”.

All these typo were substituted as requested

  • Pg 3, Line 101: Please add this reference: Carlucci, H. Xu, B. F. Scremin, C. Giannini, T. Sibillano, E. Carlino, V. Videtta, G. Gigli, G. Ciccarella, Sci. Adv. Mater. 2014, 6, 1668-1675.

The suggested reference was added in section 3.7.

  • Pg 3, Figure 1: the figure is unclear and M° in the caption is missing.

Figure 1 and the caption were clarified

  • Pg 4, Table 1: the yields and colors, indicated in the caption, are not reported.

Thanks for pointing out this mistake, the caption of the table was updated.

  • Pg 6, Figure 2: the scale is not formatted appropriately and in figures (e) and (f) is not reported. In figure (c) the caption of the axes is cut.

Figure 2 was replaced.

  • Pg 8, Figure 3: the scale is not formatted appropriately and in figure (b) is missing. In figure (f) the caption of the y axis is cut.

Figure 3 was replaced.

  • Pg 10, Figure 4: the scale is not formatted appropriately.

Figure 4 was modified.

  • Pg 11, 12, Figure 5: the scale is not formatted appropriately.

Figure 5 was modified.

  • Pg 13, Figure 6: the graphics seem to be pasted as an image and the resolution is low.

The resolution was updated.

  • References: please check carefully the numbering of the references (for example pg 12, lines 344-346).

The number of the references was updated.

  • SI, Pg 7, Figure 5: the figure (B) is cut.

This figure was corrected

Reviewer 3 Report

The manuscript detailly presents fabrication of composition with NiO nanoparticles and SiO2, TiO2, Al2O3 and Na4.2Ca2.8(Si6O18) matrices using solid-state method. The authors demonstrated that simple and environmentally friendly process was employed for the fabrication. The authors detailly studied about performance of photocatalytic properties by the degradation of blue methylene. Therefore, it is suitable for publication in Nanomaterials. However, before publication, the whole sentence should be revised to be structured. Particularly, major corrections are needed in the intro and results. In addition, each picture needs to be readjusted a little more carefully so that the reader can easily understand it. Each caption should also be checked and corrected to match what is expressed in the text.

Author Response

The manuscript detailly presents fabrication of composition with NiO nanoparticles and SiO2, TiO2, Al2O3 and Na4.2Ca2.8(Si6O18) matrices using solid-state method. The authors demonstrated that simple and environmentally friendly process was employed for the fabrication. The authors detailly studied about performance of photocatalytic properties by the degradation of blue methylene. Therefore, it is suitable for publication in Nanomaterials. However, before publication, the whole sentence should be revised to be structured. Particularly, major corrections are needed in the intro and results. In addition, each picture needs to be readjusted a little more carefully so that the reader can easily understand it. Each caption should also be checked and corrected to match what is expressed in the text.

According with the suggestions of the referee, all the text, especially the introduction and the discussion was rewritten for a better understanding. Figures were also improved.

Round 2

Reviewer 1 Report

No further revision is needed for the acceptance.

Reviewer 2 Report

The authors have carefully replied to all the suggestions and improved the quality of the work accordingly. For this reason the manuscript is suitable for pubblication in its present form.

Reviewer 3 Report

I'm satisified with revised manuscript.